# The Costs of Introducing the Hepatitis B Birth Dose Vaccine into the National Immunization Programme in Senegal (NéoVac Study)

**DOI:** 10.3390/vaccines9050521

**Published:** 2021-05-18

**Authors:** Andréa Gosset, Marie Libérée Nishimwe, Mamadou Yaya Diallo, Lucas Deroo, Aldiouma Diallo, El Hadji Ba, Patrizia Maria Carrieri, Cheikh Sokhna, Muriel Vray, Yusuke Shimakawa, Sylvie Boyer

**Affiliations:** 1Aix Marseille Univ, INSERM, IRD, SESSTIM, Sciences Economiques & Sociales de la Santé & Traitement de l’Information Médicale, ISSPAM, 13385 Marseille, France; andrea.gosset@inserm.fr (A.G.); marie.nishimwe@inserm.fr (M.L.N.); mayadiallo2008@yahoo.fr (M.Y.D.); pmcarrieri@aol.com (P.M.C.); 2Unité d’Epidémiologie des Maladies Emergentes, Institut Pasteur, 75015 Paris, France; lucas.deroo@hec.ca (L.D.); muriel.vray@pasteur.fr (M.V.); yusuke.shimakawa@pasteur.fr (Y.S.); 3VITROME, Vecteurs-Infections Tropicales et Méditerranées, Campus IRD-UCAD de Hann, Dakar CP 18524, Senegal; aldiouma.diallo@ird.fr (A.D.); el-hadj.ba@ird.fr (E.H.B.); Cheikh.Sokhna@ird.fr (C.S.); 4VITROME, Aix Marseille Univ, IRD, SSAS, IHU-MI, AP-HM, 13385 Marseille, France; 5Unité d’Epidémiologie des Maladies Infectieuses, Institut Pasteur, Dakar BP 220, Senegal; 6INSERM, 75013 Paris, France

**Keywords:** hepatitis B vaccine, birth dose, mother-to-child transmission, costs, micro-costing, Senegal, Africa

## Abstract

Some African countries are still reluctant to introduce the hepatitis B vaccine birth dose (HepB-BD) into their expanded program of immunization (EPI), partly because of logistical, economic, and cost information constraints. To assist decision-makers in these countries, we assessed the economic and financial costs of HepB-BD introduction in Senegal in 2016. We performed a micro-costing study in a representative sample of Senegal’s EPI sites at all levels in 2018. Information on EPI and HepB-BD activity-related inputs and costs was collected using standardized questionnaires and semi-structured interviews. Using inverse probability weighting, we computed weighted average costs associated with HepB-BD introduction for each EPI level, country-level aggregated costs and estimated costs per newborn. Economic and financial costs from a government perspective were estimated in US dollars for 2015, 2016 and 2017. Total economic costs were USD 143,364 in 2015, USD 759,406 in 2016 and USD 867,311 in 2017, while financial costs were USD 127,745, USD 82,519 and USD 29,853, respectively. When annualizing pre-introduction and initial training costs, the economic (financial) cost per vaccinated newborn was USD 2.10 (USD 0.30) in 2016 and USD 1.90 (USD 0.20) in 2017. Our estimates provide valuable information to implement HepB-BD in Sub-Saharan African countries that have not yet integrated this vaccine.

## 1. Introduction

Chronic hepatitis B virus (HBV) infection is a major cause of chronic liver disease and mortality in Sub-Saharan Africa [1]. Although all African countries have introduced the three-dose infant HBV vaccine (HepB3) into their current expanded program of immunization (EPI) as part of a pentavalent vaccine, HBV prevalence is still high in the region (at least 8% in adults and 3% in children under 5 years old) [1,2].

To reduce the HBV burden in Sub-Saharan Africa, priority must be given to preventing the infection in newborns and young children for the following reasons: (i) HBV transmission mainly occurs during early childhood through perinatal and horizontal transmission, (ii) the risk of developing chronic HBV infection is very high when infection occurs during the first year of life (90%) and then rapidly decreases with age to <5% in adults [3,4], (iii) the risk of developing cirrhosis and hepatocellular carcinoma (HCC) is high when chronic HBV infection occurs during early childhood, particularly through perinatal transmission [5,6].

Vaccination is the cornerstone of HBV transmission prevention. To effectively prevent perinatal and early horizontal transmission, the current HBV vaccination schedule recommended by the World Health Organization (WHO) is to administer the first dose within 24 h of birth (hepatitis B birth dose vaccine (HepB-BD)), followed by two or three doses during infancy (HepB3) [7].

In 2015, the World Health Assembly adopted a strategy to eliminate viral hepatitis as a major public health threat. Accordingly, the WHO set the goal of achieving an HBV prevalence of 0.1% in children under 5 years of age by 2030. This will require 90% and 80% HepB3 and HepB-BD coverage, respectively, in this population by that time [3]. Although these targets for elimination and vaccination coverage have been endorsed by most Sub-Saharan African countries [8], HBV vaccine coverage remains suboptimal in this region, especially for the HepB-BD. Indeed, coverage for the whole WHO African region was estimated at only 4% in 2018 [9]. This can be explained at two levels. First, only 13 countries in the region had adopted HepB-BD by 2020. Many of the rest are still reluctant to introduce it because of limited financial and technical resources. These countries also face continued challenges to prioritize resource allocation for HepB-BD and advocate for action and investment in this vaccination strategy because of the lack of reliable information on the economic and financial costs required to integrate HepB-BD into their current EPI. Second, even in countries that have already introduced the HepB-BD into their EPI, coverage of timely HepB-BD administration remains low because of the logistical difficulties in administering the dose as scheduled (i.e., within 24 h of birth) to children born at home [10,11,12].

Senegal is a lower-middle-income country in West Africa with a population of approximately 16 million inhabitants and a yearly gross domestic product of USD 1447 per capita in 2019 [13]. In 2018, total health expenditure and government health expenditure per capita were estimated at USD 58.90 and USD 14.00, respectively [14]. HBV prevalence is high, estimated at 8.1% in the general population in 2016 [2]. In 2004, Senegal’s EPI introduced the HepB3 as part of a pentavalent vaccine (diphtheria, tetanus, whooping cough, HBV, and Hemophilus influenza type B) administrated at 6, 10, and 14 weeks after birth. In 2016, the HepB-BD—using a monovalent hepatitis B vaccine—was added to the schedule. According to WHO/UNICEF, HepB3 coverage was 95% in 2017, while HepB-BD coverage was 58% in 2016 and 72% in 2017 [15].

In this context, we aimed to (i) document the experience of Senegal and the resources needed to implement timely HepB-BD; and (ii) provide accurate estimates of both economic and financial costs involved in Senegal’s introduction and implementation to help other countries plan for HepB-BD introduction in the future.

## 2. Methods

### 2.1. Study Design

This economic study was part of the ongoing Neonatal Vaccination against hepatitis B in Africa (NéoVac) program, aiming to evaluate the feasibility of introducing HepB-BD in three African countries (Burkina Faso/Senegal/Madagascar), and its impact once introduced. The program’s protocol was approved by the Clinical Research Committee of the Pasteur Institute in France (no. 2015-069) and the National Ethics Committee in Senegal (SEN18/45).

A micro-costing study was conducted in 2018 to document the quantities and costs of all resources used in Senegal in 2015, 2016 and 2017 for all its EPI activities, including introducing HepB-BD. In line with the country’s healthcare system organization, data were collected at each of the following four levels: national, regional, district and first-level health facilities. The latter include health posts (first contact facilities managed by nurses) and health centers (reference facilities at the district level where at least one medical doctor is present). Both health centers and health posts are designated as the primary sites for child immunization. The study was designed to select a representative sample of sites, which were involved in the EPI implementation in the country, as well as to measure variation in costs across the sites.

Following the sampling method proposed by Brenzel (2013) [16], we first selected five regions among Senegal’s 14 regions to represent not only the demographic, geographical and economic diversity of the country but also the different HepB-BD coverages observed. The five selected regions were Dakar, Fatick, Kedougou, Saint Louis and Zinguichor. Combined, they cover 6,702,015 inhabitants or 40% of the total population. In each of the five selected regions, we then selected three districts with different HepB-BD coverages. Within each selected district, we randomly selected six primary care facilities (health centers and health posts). We also included two agencies at the national level, specifically the Prevention Department of the Ministry of Health, which is in charge of the EPI at the national level, and the National Medical Supplies Agency, where all the EPI vaccines are stored before being dispatched to regional offices.

The target sample, therefore, included 112 sites, specifically, the 2 agencies at the national level, 5 regional health offices, 15 district health offices within those regions, and 90 primary care facilities within those districts.

### 2.2. Data Collection

Data collection for the micro-costing study took place between March and October 2018 in 100 over the 112 target sample sites. We were not able to visit the 12 healthcare facilities initially selected due to the unavailability of staff.

Semi-structured interviews were performed with key informants and standardized data collection forms. We collected detailed information on activities implemented for HepB-BD introduction and EPI management, the corresponding inputs used, and associated costs at the program’s four levels. Activities and resources used for implementing HepB-BD and managing the national EPI were identified using a method proposed by the WHO and Brenzel (2013) [16,17].

Specifically, interviews were performed with managers of the Prevention Department of the Ministry of Health, the National Medical Supplies Agency, the regional health offices and district health offices, as well as with EPI managers at the different levels of the program. Collected information included details on the organization of EPI activities (vaccine administration, distribution and storage, program management and supervision), resources used (including human resources and time spent), as well as details on activities specific to introducing HepB-BD (initial training, social mobilization, additional cold chain and distribution equipment) and resources associated with these activities. Semi-structured interviews were also conducted with managers of primary care facilities and staff involved in maternal and child health services to document vaccination-related organization (routine and outreach) and resource use, including the time spent on HepB-BD immunization-related activities.

Besides the semi-structured interviews, forms were filled in at each study site to collect quantitative information on resources used for the EPI and HepB-BD activities (type, quantities, and unit cost).

Data were entered into an Excel spreadsheet and cross-checked by two investigators (M.Y.D. and M.L.N.).

### 2.3. Costs Analysis

#### 2.3.1. Overview of the Costing Analysis

Data obtained in the 2018 micro-costing study were used to estimate the incremental economic and financial costs associated with the introduction and implementation of HepB-BD in Senegal. Economic costs included a valuation of all inputs needed for the introduction and routine implementation of HepB-BD, including valuation of time, supplies and equipment already existing in the EPI and used for HepB-BD (as the use of these resources represents an opportunity cost for the EPI) [16,17]. Financial costs were defined in terms of how much money was paid for the additional resources used for HepB-BD introduction and implementation [16]. Accordingly, no financial cost was considered for activities carried out as part of the overall implementation of the EPI (such as vaccine storage and distribution, program supervision and management, and waste management) unless additional resources were specifically required for HepB-BD introduction or implementation. Furthermore, capital costs were annualized over their expected lifetime using a discount rate of 3% in the economic approach, and a straight-line depreciation was used in the financial approach.

Annual economic and financial costs were estimated using a government perspective in 2016 US dollars for 2015 (year of HepB-BD pre-introduction and initial training on its administration), 2016 (year of HepB-BD introduction) and 2017 (year of routine HepB-BD implementation).

Using the inverse probability weighting method (see Section 2.3.3. below), we estimated the weighted average economic and financial costs associated with HepB-BD implementation for the regional, district and primary care levels of the EPI. Weighted average costs were multiplied by the corresponding number of sites to generate the total cost for each level of EPI. Costs obtained for each level of the EPI were then aggregated with the costs of the national level to obtain the total cost at the country level. The costs of vaccines and safety supplies were directly estimated for the whole country and reported nationally. Finally, we estimated the financial and economic costs of HepB-BD vaccination per newborn.

#### 2.3.2. Costing Methods to Estimate Costs per Study Site

Activities and input line items included in the costing analysis are described in Table 1. Additional information on the main characteristics of Senegal and the data used in the costing analysis is presented in Table 2.

Resources were classified into the following activities: pre-introduction (development of the national plan for HepB-BD introduction, development of associated training tools, updating and printing of immunization-related materials, including child immunization cards, brochures, information sheets, etc.), initial training, social mobilization, HepB-BD vaccine administration, storage, distribution, supervision, EPI management and waste management. For each activity, we distinguished between recurrent costs (labor, per diems and transport allowances, vaccines and consumables, cold chain energy and maintenance, fuel for transportation and maintenance of vehicles, printing and other operating costs) and capital costs (vehicles, cold chain equipment, incinerators and IT equipment).

Pre-introduction, initial training, social mobilization and HepB-BD vaccine administration were specific to the introduction and implementation of HepB-BD. Accordingly, their whole cost was attributed to HepB-BD both in the economic and financial approaches (financial costs did not include labor time as no additional staff were hired). The rest of the activities were related to implementing the EPI as a whole. To estimate economic costs, we first estimated the EPI costs associated with each of these activities and then allocated a share of this cost to the HepB-BD implementation using the allocation criteria (Table 1). To estimate financial costs, we only considered expenditures related to the purchase of resources specifically required for HepB-BD introduction or implementation.

We estimated economic and financial costs for 2015, 2016 and 2017. Unit costs in CFA franc (CFAF) and in Euros were converted to US dollars (USD), using the World Bank exchange rate for 2016 (1 USD = 592,542 FCFA [18]). For each study site and for each activity, the costs were computed by multiplying the resources used by their unit costs. The total cost was then obtained by summing the costs of the different activities.

#### 2.3.3. Costs Aggregation to Estimate the Total Annual Costs at the Country Level

We used inverse probability weighting to estimate annual average weighted costs for the regional, district and first-level health facility levels. We then aggregated costs at the country level (15). Weights were computed for each study site as the inverse probability of being selected. The respective weighted average costs obtained for each level of the EPI, excluding the national level, were multiplied by the number of sites at the corresponding level to obtain the total annual cost at each level of the EPI for 2015, 2016 and 2017. Finally, the total annual cost at the country level was obtained by summing the respective total annual costs estimated at each level of the health system (i.e., national, regional, district and first-level health facility), as illustrated in Figure 1.

#### 2.3.4. Estimation of the Cost per Vaccinated Newborn

As the 2015 annual costs were related to pre-introduction activities and initial training, we considered them as initial investment (i.e., capital) costs. Accordingly, we computed the corresponding annualized costs over five and two years, respectively, as recommended by the WHO [17]. Investment annualized costs were then added to the 2016 and 2017 annual costs to obtain annual total costs at the country level, including the cost of initial investment related to pre-introduction activities and initial training.

Finally, we divided the 2016 and 2017 economic and financial total costs estimated at the country level (with and without the annualized cost of initial investment) by the total number of newborns who received the HepB-BD (irrespective of whether or not administration occurred according to the WHO recommendation (i.e., within 24 h of birth)). The latter number was estimated using the database used in the EPI for vaccines stocks management, while the number of newborns who received timely HepB-BD-as per the WHO recommendation-was estimated using data provided by the Ministry of Health of Senegal to WHO/UNICEF [19].

Analyses were conducted using R version 3.6.3 (R Core Team, Vienna, Austria).

## 3. Results

### 3.1. Introduction and Implementation of HepB-BD in Senegal

#### 3.1.1. Pre-Introduction Activities

In 2014, an HepB-BD introduction plan was developed and discussed at a workshop, which brought together national EPI staff and experts from the WHO and the United States Agency for International Development. A steering committee and several subcommittees were organized to coordinate the preparation, implementation and monitoring of the HepB-BD introduction. In 2015, several specific workshops were held to develop training tools for healthcare workers and communication materials for social mobilization. Training sessions were subsequently organized at each level of the healthcare system for a total of 2000 healthcare workers throughout the country. Social mobilization was then implemented at a national level with the support of the regional and district health offices. The purchase of monovalent HepB-BD vaccines (10-dose vials; Hepavax-Gene^®^, Berna Biotech Korea Corp., Korea) was overseen by the Prevention Department of the Ministry of Health. Vaccines were stored in the EPI cold chain at the National Medical Supplies Agency. Using existing EPI supply chains, HepB-BD vaccines were transported from the central EPI store to the regional offices, then from the latter to the district offices, and in turn from the district offices to first-level health facilities. HepB-BD vaccination started in January–February 2016.

#### 3.1.2. HepB-BD Vaccination Strategy

Given the relatively high rate of institutional childbirth (estimated at 61% in 2017 according to the Ministry of Health of Senegal), the Ministry of Health chose fixed facility-based immunization—rather than outreach immunization—as the main strategy to deliver HepB-BD. More specifically, the national strategy aimed to: (i) promote using maternal and child health services and institutional delivery through community sensitization; (ii) ask women who gave birth outside health facilities to bring their newborn(s) to a health facility as soon as possible for postnatal care, including vaccination; (iii) make monovalent hepatitis B vaccines both widely available for HepB-BD administration, and free of charge in all healthcare facilities providing childbirth, maternal, and child health services.

In addition to the fixed facility-based HepB-BD immunization, first-level health facilities also deliver HepB-BD during routine outreach vaccination activities. These are regularly (most often monthly) conducted to reach populations living more than five km away.

### 3.2. Estimations of Weighted Average Costs per Activity

Table A1 presents a descriptive analysis of the weighted average economic costs, overall and per activity, for each level of the EPI (excluding the national level). In 2016 (2017), the weighted average economic costs (excluding vaccines costs for healthcare facilities) were USD 2473 (USD 2093) per regional health office, USD 2389 (USD 2396) per district health office, USD 743 (USD 1015) per health center and USD 243 (USD 313) per health post. At the regional and district levels, the activities with the highest costs were supervision and EPI management. In contrast, at the first-level facility level, vaccine administration was the main cost driver.

### 3.3. Costs of HepB-BD Implementation at the Country Level

The total economic cost of the HepB-BD implementation in Senegal were estimated at USD 143,364 for 2015, USD 759,406 for 2016 and USD 875,240 for 2017 (Table 3). Initial training in HepB-BD administration accounted for more than half of the annual costs in 2015 (54%). In 2016 and 2017, vaccine administration (including time spent) was the main cost activity (57%, USD 435,596 in 2016; 63%, USD 556,224 in 2017), followed by supervision of EPI activities (17%, USD 126,648 in 2016; 17%, USD 147,443 in 2017), and EPI management (11%, USD 79,841 in 2016; 11%, USD 96,000 in 2017).

The total financial costs were much lower as they did not account for the opportunity costs of existing EPI resources, mainly human resources already in place and existing equipment. They were estimated at USD 127,745 for 2015, USD 81,940 for 2016 and only USD 29,915 for 2017 (Table 4). In 2015, the main cost driver was initial training (61% of the total cost). In 2016, social mobilization (74%) was the main cost activity, followed by vaccine administration (23%), while in 2017, the main cost activity was vaccine delivery (82%), followed by social mobilization (11%).

### 3.4. Cost per HepB-BD Vaccinated Newborn

In 2016 (2017), an estimated 379,999 (473,643) newborns received HepB-BD, including 314,084 (391,488) within 24 h of birth, for total incremental economic and financial costs (including the annualized cost of initial investment) of USD 809,819 (USD 924,185) and USD 130,745 (USD 78,720), respectively (Table 5).

The economic cost per vaccinated newborn (irrespective of whether or not administration occurred within 24 h of birth) were USD 2.00 in 2016 and USD 1.80 in 2017 (excluding the annualized cost of initial investment), and USD 2.10 in 2016 and USD 1.90 in 2017 (including the annualized cost of initial investment).

The economic costs per newborn vaccinated within 24 h of birth were slightly higher: USD 2.40 (2016) and USD 2.20 (2017) when excluding the annualized cost of initial investment, and USD 2.60 and USD 2.40 when including it.

When, including this cost, the financial costs per vaccinated newborn (irrespective of timing) were USD 0.30 (2016) and USD 0.20 (2017), while the financial costs per newborn vaccinated within 24 h of birth were USD 0.40 (2016) and USD 0.20 (2017).

## 4. Discussion

To our knowledge, this is the first study to document both economic and financial costs associated with introducing the HepB-BD in Sub-Saharan Africa using real-world data. We investigated the experience of Senegal, a country, which successfully introduced this vaccine dose in 2016. Using data from a micro-costing study designed to obtain a representative sample of sites involved in implementing the country’s EPI, we identified and valued the resources mobilized for a large range of activities related to the introduction and implementation of the HepB-BD at the different levels of Senegal’s health system. The economic costs per HepB-BD vaccinated newborn ranged from USD 2.00 and 2.60 for the first year of introduction (2016) and from USD 1.80 and 2.40 for the subsequent year (2017). Corresponding financial costs were 6 to 11 times lower (ranging from USD 0.20–0.40), mainly because they did not consider the time spent by human resources on pre-introduction activities, HepB-BD administration, EPI management and supervision.

Costs per HepB-BD vaccinated newborn in the current analysis were higher than those reported in a previous study, which we conducted in 2017 in the Dafra district of Burkina Faso using a hypothetical approach (USD 2.94 for the first year of introduction and USD 0.54 for the subsequent year) [20]. However, these differences must be interpreted with care given the different methodologies used. The Burkinabe study was hypothetical and used assumptions about resources used, as the country has not yet introduced the HepB-BD into its EPI. In the present study on Senegal, we worked with real-world data collected in the 2018 micro-costing study. Furthermore, unlike the present study, we did not annualize the initial investment costs related to the initial training and pre-introduction activities in the Burkinabe study. This may explain the high (hypothetical) cost per vaccinated newborn found for the first year in Burkina Faso. In addition, EPI management and supervision costs, which accounted for a substantial proportion of the economic cost in the present study, were not considered in our Burkinabe study.

Furthermore, our cost estimates for Senegal were similar to that Brenzel et al. found in Benin (2015). They reported a cost of USD 2.00 per vaccine dose administrated in that country’s EPI (considering all vaccines administrated in the EPI; Benin had not introduced HepB-BD at that time) [21]. This would suggest that the HepB-BD has an economic cost similar to other vaccines currently implemented in EPI in West African countries.

The strategy adopted by the Senegalese health authorities was to introduce the HepB-BD as part of the existing routine facility-based vaccination program. Institutional childbirth was particularly promoted to pregnant women. When this was not possible, they were asked to bring their newborn(s) to a healthcare facility for postnatal care and to receive a birth certificate as soon as possible. However, as no clear recommendation was made about whether or not to include HepB-BD in outreach immunization strategy, first-level health facilities implemented different associated practices: some used this strategy to provide HepB-BD to unvaccinated newborns as a catch-up dose. While they may have helped reach some additional newborns, these strategies most likely missed out on vaccinating newborns within the recommended first 24 h after birth and consequently may have not effectively protected newborns against the risk of perinatal infection. Specific studies are needed to evaluate the effectiveness of outreach HepB-BD strategies in Sub-Saharan Africa.

The experience of Senegal described in this work will be useful for other African countries where the HepB-BD has not yet been introduced. This may also be relevant to Gavi, the Vaccine Alliance partnership, which expressed its future support to help countries administer the HepB-BD according to the WHO-recommend schedule [22].

Even after the introduction of HepB-BD into a country’s EPI, considerable challenges remain to reach a high level of coverage in the resource-limited African context. First, a recent meta-analysis of HepB-BD coverage in Sub-Saharan African countries where the dose is scheduled in the EPI reported that only 0.3% of babies born in healthcare facilities received the dose within 24 h of birth [11]. This highlights the importance of both providing clear guidelines for timely HepB-BD administration in healthcare facilities and ensuring the availability of the HepB-BD at all levels of the healthcare system [12,23]. As illustrated by the Senegalese experience, monovalent hepatitis B vaccines must be made available in hospitals’ maternity wards. They often do not have a routine EPI vaccination activity. Second, only 60% of women give birth in healthcare facilities in Sub-Saharan Africa [24]. Accordingly, even if all infants born in healthcare facilities were to receive the HepB-BD according to the WHO recommendation, achieving the target of 80% HepB-BD coverage by 2030 would be difficult [3]. Additional strategies to reach babies not born in healthcare facilities are urgently needed in Sub-Saharan Africa, including the systematic notification of home births to EPI staff by lay community workers [25], increased home-based outreach vaccination using mono-dose vaccines (either in vials or in pre-filled auto-disable injection devices (cPAD)) [26,27], and ambient storage of HepB-BD under a controlled temperature chain [27].

The main limitation of the present study concerns the uncertainty surrounding the definition of allocation criteria used in the analysis of the economic costs to allocate shared costs (i.e., costs associated with activities not specifically carried out for the HepB-BD introduction but, which are carried out as part of the overall implementation of the EPI, and therefore, are of benefit for all EPI vaccines, including the HepB-BD). This is especially the case for EPI management and supervision as well as vaccine storage and distribution. Allocation criteria for labor costs were based on labor time reported by staff, who were asked to both determine the time they spent on the EPI and then, within the EPI, the time spent on the HepB-BD administration component. Because of our exclusive focus on HBV immunization and HepB-BD, time allocations may have been overestimated relative to other services. Consequently, they may have led to an overestimation of labor and economic costs. Another important piece of information used to determine the allocation of shared costs in our study regarded vaccine storage and distribution is the number of vaccine doses delivered as part of the EPI (for the HepB-BD and other vaccines). Although we obtained this information from the database developed by WHO and used in the Senegalese EPI for vaccine stock management, we cannot completely exclude the risk of errors in the source.

Another study limitation regards the estimated number of newborns receiving the HepB-BD dose within the recommended 24 h after birth. As this information was not available in the database used for the EPI vaccine stock management, we used estimated data provided by the Ministry of Health of Senegal for a WHO/UNICEF report [19].

## 5. Conclusions

Our costs analyses based on the Senegal context provide important insights for other African countries regarding the types and amount of resources and costs required to introduce HepB-BD. They can help policymakers and other key stakeholders design an effective HepB-BD-introduction program, estimate program-related costs, and develop vaccination strategies in resource-limited contexts. Furthermore, our estimations of the economic costs in Senegal will also be especially useful to estimate and confirm findings from two studies suggesting the cost-effectiveness of introducing HepB-BD into Sub-Saharan African EPI [20,28]. Further studies evaluating the effectiveness of complementary strategies to improve HepB-BD coverage and its timely administration are urgently needed in Sub-Saharan Africa.

## Figures and Tables

**Figure 1 vaccines-09-00521-f001:**
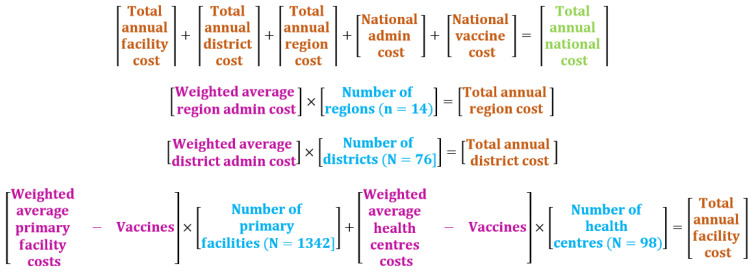
Aggregation method for calculating costs of HepB-BD implementation in Senegal. Source: Adapted from Brenzel L. Common approach for the costing and financing analyses of routine immunization and new vaccine introduction costs (NUVI). Bill and Melinda Gates Foundation; 2013.

**Table 1 vaccines-09-00521-t001:** Summary of the activities and input line items considered in the costing analysis covering the 2015–2017 period in Senegal.

Activity	Description	Activity Related to HepB-BD Introduction/Implementation	Input Line Items	Level of the Health System	Comments
**1. Pre-introduction**	Activities carried out to prepare for HepB-BD introduction, including developing the national plan for the introduction, developing training tools needed, updating and printing immunization-related materials (child immunization cards, registers, EPI supervision tools, etc.)	Yes	Recurrent costs: time of salaried labor, per diems, fuel for transportation, stationary, room rental, printing (immunization cards and other immunization related materials)	National	Costs mainly incurred in 2015 and considered as investment costs (annualized over 5 years) Financial costs do not include labor time as no additional staff were hired
**2. Initial training**	Initial training and supervision specifically for HepB-BD introduction	Yes	Recurrent costs: staff (provision of services), per diems and transport allowances, fuel for transportation, stationary, room rental	National, regional and district	Costs incurred in 2015 and considered as investment costs (annualized over 2 years)
**3. Social mobilization**	Social mobilization associated with HepB-BD introduction (during the introduction period and after): development of radio publicity spots and communication tools, broadcast of radio spots and TV shows, organization of special events	Yes	Recurrent: staff (provision of services), per diems and transport allowances, fuel for transportation, room rental for special events, radio time and TV shows to broadcast messages, stationery, printing communication tools (flyers, posters, etc.)	All (national, regional, district and primary care facility)	Costs incurred mainly in 2016 (a small number of costs are reported in 2017)
**4. Vaccine administration**	Routine facility-based administration of HepB-BD vaccine and outreach vaccination	Yes	Recurrent costs: time taken for a salaried worker to administer the HepB-BD vaccine (6 min per infant), vaccines (number of doses administered), vaccine injection and safety supplies (syringes and safety boxes, etc.), per diems and fuel for outreach vaccination Capital costs: vehicles	National level for vaccines and supplies costs District and first-level health facility levels for labor time related to vaccine administration	Costs incurred in 2016 and 2017 Financial costs do not include labor time as no additional staff were hired
**5. Vaccine storage**	Cold chain storage of vaccines and storage of injection and safety supplies	No (EPI costs allocated to the HepB-BD based on the share of the HepB-BD volume relative to the total volume for all current vaccines	Recurrent costs: cold chain maintenance and cold chain energy (electricity, gas, ice packs, etc.) Capital costs: cold chain equipment	All (national, regional, district and first-level health facility)	Costs incurred in 2016 and 2017 Financial costs only include annualized costs of new cold chain equipment purchased in 2015 and 2016
**6. Vaccine distribution**	Transportation of vaccines, injection and safety supplies	No (EPI costs allocated to the HepB-BD based on the share of the HepB-BD volume relative to the total volume for all current vaccines	Recurrent costs: per diems, fuel for transportation of vaccines and supplies, vehicle maintenance Capital costs: vehicles	All (national, regional, district and first-level health facility)	Costs incurred in 2016 and 2017 Financial costs only include the annualized costs of new vehicles purchased in 2015 and 2016
**7. Supervision, monitoring and surveillance**	Routine supervision of immunization activities, monitoring and evaluation of immunization data (including surveillance of post-vaccination events)	No: Costs–excluding salary costs-allocated to HepB-BD based on the share of the number of doses for the HepB-BD relative to the total number of vaccines doses in the EPI Salary costs allocated based on time spent	Recurrent costs: time of salaried labor, per diems and transport allowances, fuel for transportation, vehicle maintenance Capital costs: vehicles	All (national, regional, district and first-level health facility)	Costs incurred in 2016 and 2017 No financial costs
**8. Management**	Planning, budgeting and managing the immunization program (including continuing training)	No: costs-excluding salary costs-allocated to the HepB-BD based on the share of the number of doses for the HepB-BD relative to the total number of vaccines doses in the EPI Salary costs allocated based on time spent	Recurrent costs: time of salaried labor, per diems and transport allowances, fuel for transportation Capital costs: office equipment (computers, printers, etc.)	All (national, regional, district and first-level health facility)	Costs incurred in 2016 and 2017 No financial costs
**9. Waste management**	Management of HepB-BD vaccine-related waste	No (costs allocated to the HepB-BD based on the share of the HepB-BD volume relative to the total volume for all current vaccines	Recurrent costs: incinerator fuel Capital costs: incinerator	Regional and district levels	Costs incurred in 2016 and 2017 No financial costs

Abbreviations: EPI = expanded program on immunization; HepB-BD = hepatitis B birth dose.

**Table 2 vaccines-09-00521-t002:** Main data used in the costing analysis of hepatitis B birth dose introduction in Senegal.

Main Data	Description	Source
Target population (live births)	541,529 in 2016; 543,126 in 2017	World Bank
HepB-BD coverage rates	58% in 2016; 72% in 2017	WHO/UNICEF
Total number of HepB-BD administered doses	379,999 in 2016; 473,643 in 2017	National database (DVD-MT) used for the management of vaccine stocks in the EPI
Total number of HepB-BD administered doses within 24 h after birth	314,084 in 2016; 391,488 in 2017	WHO/UNICEF
Vaccine type	10-dose vials (Hepavax-Gene^®^, Berna Biotech Korea Corp., Incheon, Korea)	Ministry of Health of Senegal
Vaccine wastage rates	5%	Ministry of Health of Senegal
Vaccine unit price	USD 0.20	UNICEF
Vaccine administration	Within 24 h after birth	Ministry of Health of Senegal
Structures where HepB-BD is delivered	Primary care facilities (public and subsidized private) Maternity ward in health centers Maternity ward in regional hospitals	Ministry of Health of Senegal
Personnel in charge of HepB-BD administration	Nurses	
Time needed to administer HepB-BD	6 min	NéoVac study
Discount rate for capital costs	3%	WHO

Abbreviations: EPI = expanded program on immunization; HepB-BD = hepatitis B birth dose; UNICEF = United Nations Children’s Fund; WHO = World Health Organization.

**Table 3 vaccines-09-00521-t003:** Aggregated economic costs (in USD 2016) per activity and per study site.

		National			Regional Health Offices			District Health Offices			Health Posts		Health Centers		Total			% per Activity		
		2015	2016	2017	2015	2016	2017	2015	2016	2017	2016	2017	2016	2017	2015	2016	2017	2015	2016	2017
**Pre-introduction**	Capital	0														0	0			
	Running	65,855														0	0			
	Total	65,855													65,855	0	0	45.9%	0.0%	0.0%
**Initial training**	Capital		0	0		0	0	0	0	0	0	0	0	0	0	0	0			
	Running		0	0	9272	0	0	68,238	0	0	9	0	0	0	77,510	9	0			
	Total		0	0	9272	0	0	68,238	0	0	9	0	0	0	77,510	9	0	54.1%	0.0%	0.0%
**Social mobilization**	Capital		0	0		0	0		0	0	0	0	0	0	0	0	0			
	Running		20,653	0		6143	0		31,923	357	1470	1189	688	1645	0	60,877	3191			
	Total		20,653	0		6143	0		31,923	357	1470	1189	688	1645	0	60,877	3191		8.0%	0.4%
**Vaccine administration**	Capital	0	0	0		0	0		2118	2907	17,326	23,496	0	0	0	19,444	26,402			
	Running	0	98,835	123,191		0	0		2367	3741	245,579	313,652	69,372	89,238	0	416,152	529,821			
	Total	0	98,835	123,191		0	0		4485	6647	262,904	337,147	69,372	89,238	0	435,596	556,224		57.4%	63.6%
**Storage**	Capital		404	405		2292	2292		3863	4699	15,908	19,761	5351	10,520	0	27,820	37,677			
	Running		2651	2652		928	897		1333	1550	12,466	15,824	232	313	0	17,610	21,236			
	Total		3055	3057		3220	3189		5197	6249	28,374	35,585	5583	10,833	0	45,430	58,913		6.0%	6.7%
**Distribution**	Capital		634	635		2355	2355		1984	2448	2292	3070	59	84	0	7326	8593			
	Running		701	658		9	9		40	49	2473	3577	233	327	0	3455	4620			
	Total		1335	1293		2364	2364		2024	2496	4766	6647	292	412	0	10,781	13,213	0.0%	1.4%	1.5%
**Supervision**	Capital		1585	1586		7367	7367		16,943	23,252	7309	11,694	0	0	0	33,204	43,898			
	Running		4856	1333		7360	7617		76,303	88,262	4925	6332	0	0	0	93,443	103,544			
	Total		6441	2918		14,726	14,984		93,246	111,515	12,234	18,026	0	0	0	126,648	147,443		16.7%	16.8%
**Management**	Capital		94	94		159	159		513	775	1431	1600	222	261	0	2419	2888			
	Running		2442	2442		7933	8515		63,142	73,037	3822	7293	84	1824	0	77,422	93,111			
	Total		2535	2536		8092	8675		63,654	73,812	5252	8892	306	2085	0	79,841	96,000		10.5%	11.1%
**Waste management**	Capital					94	94		0	0	0	0	0	0	0	94	94			
	Running					0	0		132	162	0	0	0	0	0	132	162			
	Total					94	94		132	162	0	0	0	0	0	226	256	0.0%	0.0%	0.0%
**Total**		65,855	132,854	132,995	9272	34,640	29,306	68,238	200,660	201,239	315,010	407,487	76,241	104,213	143,364	759,406	875,240	100.0%	100.0%	100.0%
**% per site type**		45.9%	17.5%	15.2%	6.5%	4.6%	3.3%	47.6%	26.4%	23.0%	41.5%	46.6%	10.0%	11.9%	100.0%	100.0%	100.0%			

**Table 4 vaccines-09-00521-t004:** Aggregated financial costs (in USD 2016) per activity and per study site.

		National			Regional Health Offices			District Health Offices			Health Posts		Health Centers		Total			% per Activity		
		2015	2016	2017	2015	2016	2017	2015	2016	2017	2016	2017	2016	2017	2015	2016	2017	2015	2016	2017
**Pre-introduction**	Capital	0													0	0	0			
	Running	50,226													50,226	0	0			
	Total	50,226													50,226	0	0	39.3%		
**Initial training**	Capital		0	0		0	0	0	0	0	0	0	0	0	0					
	Running		0	0	9272	0	0	68,238	0	0	9	0	0	0	77,519					
	Total		0	0	9272	0	0	68,238	0	0	9	0	0	0	77,519			60.7%		
**Social mobilization**	Capital		0	0		0	0	0	0	0	0	0	0	0	0	0	0			
	Running		20,653	0		6143	0	0	31,923	357	1470	1189	688	1645	0	60,877	3192			
	Total		20,653	0	0	6143	0	0	31,923	357	1470	1189	688	1645	0	60,877	3192		74.3%	10.7%
**Vaccine administration**	Capital		0	0	0	0	0	0	0	0	0		0	0	0	0	0			
	Running		0	0	0	0	0	0	2367	3741	16,829	20,875	0	0	0	19,195	24,616			
	Total		0	0	0	0	0	0	2367	3741	16,829	20,875	0	0		19,195	24,616		23.4%	82.3%
**Storage**	Capital		152	152	0	182	182	0	197	226	159	161	799	928	0	1490	1649			
	Running		0	0	0	0	0	0	0	0	0	0	0	0	0	0	0			
	Total		152	152	0	182	182	0	197	226	159	161	799	928	0	1490	1649		1.8%	5.5%
**Distribution**	Capital		0	0	0	0	0	0	362	439	14	17	2	2	0	378	458			
	Running		0	0		0	0	0	0	0	0	0	0	0	0	0	0			
	Total		0	0	0	0	0	0	362	439	14	17	2	2	0	378	458		0.5%	1.5%
**Supervision**	Capital		0	0	0	0	0	0	0	0	0	0	0	0	0	0	0			
	Running		0	0	0	0	0	0	0	0	0	0	0	0	0	0	0			
	Total		0	0	0	0	0	0	0	0	0	0	0	0	0	0	0			
**Management**	Capital		0	0	0	0	0	0	0	0	0	0	0	0	0	0	0			
	Running		0	0	0	0	0	0	0	0	0	0	0	0	0	0	0			
	Total		0	0	0	0	0	0	0	0	0	0	0	0	0	0	0			
**Waste management**	Capital		0	0	0	0	0	0	0	0	0	0	0	0	0	0	0			
	Running		0	0	0	0	0	0	0	0	0	0	0	0	0	0	0			
	Total		0	0	0	0	0	0	0	0	0	0	0	0	0	0	0			
**Total**		**50,226**	**20,804**	**152**	**9272**	**6325**	**182**	**68,238**	**34,849**	**4763**	**18,481**	**22,243**	**1490**	**2575**	**127,745**	**81,940**	**29,915**	**100%**	**100%**	**100%**
% site type		39.3%	25.4%	0.5%	7.3%	7.7%	0.6%	53.4%	42.5%	15.9%	22.6%	74.4%	1.8%	8.6%	100%	100%	100%			

**Table 5 vaccines-09-00521-t005:** Economic and financial costs (in USD 2016) per HepB-BD vaccinated newborn in Senegal.

		Total Annual Economic Cost	Total Annual Financial Cost	Economic Cost/Vaccinated Children (Irrespective of Timing)	Economic Cost/Vaccinated Children within 24 h	Financial Cost/Vaccinated Children (Irrespective of Timing)	Financial Cost/Vaccinated Children within 24 h
Costs, excluding the annualized costs of initial investment	2016	759 406	81 940	2.00	2.42	0.22	0.26
	2017	875 240	29 915	1.85	2.24	0.06	<0.01
Costs, including the annualized costs of initial investment	2016	809 819	130 745	2.13	2.58	0.34	0.42
	2017	924 185	78 720	1.95	2.36	0.17	0.20

## Data Availability

Data available on request due to privacy/ethical reasons.

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
