# Peer review of "The Costs of Introducing the Hepatitis B Birth Dose Vaccine into the National Immunization Programme in Senegal (NéoVac Study)"

_vaccines, 2021, doi:10.3390/vaccines9050521_

Round 1

Reviewer 1 Report

Hepatitis B virus (HBV) remains an important cause of morbidity and mortality on the African continent and elsewhere in the world. The fact that it is mainly transmitted in during early childhood through perinatal and horizontal routes and its potential to cause chronic liver disease and hepatocellular carcinoma when contracted during the first year of life, makes it a significant health issue. Sub-Saharan Africa still experiences a relatively high burden of HBV infection in the adult and pediatric populations, adding greater urgency to this issue. Vaccination has proven extremely successful in the developed parts of the world, significantly reducing the burden of early childhood infection with HBV. The WHO currently recommends a three-dose vaccination schedule for HBV and this includes birth dose within 24 hours of birth. In developed countries like the US, every baby is given this hepatitis B birth dose (HepB-BD), prior to leaving the hospital. In order for the WHO and these countries to achieve the goal of reducing the prevalence of HBV infection in children under five years old to 0.1% by 2030, there will need to be significant advances in increasing HepB-BD across the region. Since cost is a major impediment to ramping up to the EPI standard, having a systematic cost analysis for governments and decision makers is an important part of the formula for success.

The current manuscript is a micro-costing analysis of a representative country in the region, exploring all levels of the vaccine distribution and delivery network for the Expanded Program of Immunization (EPI) in 2018. This is a well-written manuscript by Gosset, et al, exploring the economic and financial cost of introducing the Hepatitis B vaccine birth dose into the Expanded Program of Immunization in Senegal, concludes that each vaccine dose will cost the government just over US $2.00 in economic cost based on data from 2015-2017. The financial cost was between US $0.20 and US $0.30.  This cost analysis is an important reference for Senegal as well as other Sub-Saharan African countries with similar economic constraints. The authors have systematically broken down the costs for each level of the system as well as determined that the greatest expense came from personnel costs to administer the vaccine.

Minor questions

In table 1 you have capital cost for vehicles. Does this assume the acquisition of new vehicles or maintenance of current vehicles?

In table 1 there is a capital cost for incinerators. This gives the impression that new incinerators would be needed because of the addition of this single HepB-BD dose. Would the country not have incinerators in place already to deal with other medical waste disposals?

Figure 1. The first formula [Total annual facility cost] [total annual district cost] – [Total annual regional cost] + [National Admin Cost] + [National vaccine cost] = [Total annual national cost] is used. This is a bit confusing to me and did not seem to align with the formula in the document cited (http://static1.squarespace.com/static/556deb8ee4b08a534b8360e7/t/55970258e4b03cf942da51ac/1435959896232/WEBSITE_Common+Approach.pdf).

The authors should double check this formula. It is possible that I am just missing something here about these aggregation formula methods.  

Author Response

Minor questions

In table 1 you have capital cost for vehicles. Does this assume the acquisition of new vehicles or maintenance of current vehicles?

Maintenance costs of vehicles are considered as recurrent costs. Capital costs for vehicles are thus only related to the acquisition of vehicles. The classification of input line items between capital and recurrent costs is described in section 2.3.2. p6:

“For each activity, we distinguished between recurrent costs (labour, per diems and transport allowances, vaccines and consumables, cold chain energy and maintenance, fuel for transportation and maintenance of vehicles, printing and other operating costs) and capital costs (vehicles, cold chain equipment, incinerators and IT equipment).”

The same classification is applied in Table 1 where the maintenance of vehicles is listed as “recurrent cost”. In addition, we would like to clarify that for the activities which are non-specifically related to the HepB-BD introduction or implementation (as indicated in the 3rd column of Table 1), economic and financial costs related to equipment (such as vehicles) are not estimated using the same approach:

  • For the economic costs: we considered both the opportunity costs of existing vehicles already in place in the EPI and the costs of the new vehicles acquired for the HepB-BD introduction.
  • For the financial costs: we only considered the costs of the new vehicles specifically acquired for the HepB-BD introduction.

This has been specified in the last column “comments” of the Table 1 (See line 6. Vaccines distribution) as well as in the third paragraph of the subsection 2.3.2. p6-7:

“Pre-introduction, initial training, social mobilization and HepB-BD vaccine administration were specific to the introduction and implementation of HepB-BD. Accordingly, their whole cost was attributed to HepB-BD both in the economic and financial approach (but the financial costs do not include labour time as no additional staff were hired). The rest of the activities were related to the implementation of the EPI as a whole. To estimate economic costs, we first estimated the EPI costs associated with each of these activities and then allocated a share of this cost to the HepB-BD implementation using the allocation criteria (Table 1). To estimate financial costs, we only considered expenditures related to the purchase of resources specifically required for HepB-BD introduction or implementation.”

In table 1, there is a capital cost for incinerators. This gives the impression that new incinerators would be needed because of the addition of this single HepB-BD dose. Would the country not have incinerators in place already to deal with other medical waste disposals?

Indeed, the country already had incinerators to deal with other medical waste disposals (mainly EPI vaccines). However, this equipment is also used for the HepB-BD. This is why, in the economic approach, a share of the cost of incinerators is allocated to the HepB-BD (based on the share of the HepB-BD volume relative to the total volume for all current vaccines). This cost represents the opportunity cost of using the equipment for the HepB-BD. In the financial approach, we only consider the cost of new equipment purchased for HepB-BD. As no incinerators have been purchased for the HepB-BD introduction, there is no financial cost. This is indicated in the last column “comments” of the Table 1 (See line 9. Waste management).

Figure 1. The first formula [Total annual facility cost] [total annual district cost] – [Total annual regional cost] + [National Admin Cost] + [National vaccine cost] = [Total annual national cost] is used. This is a bit confusing to me and did not seem to align with the formula in the document cited (http://static1.squarespace.com/static/556deb8ee4b08a534b8360e7/t/55970258e4b03cf942da51ac/1435959896232/WEBSITE_Common+Approach.pdf).

The authors should double check this formula. It is possible that I am just missing something here about these aggregation formula methods.  

Thank you very much for your comment. The formula was indeed incorrect due to missing “+” and a wrong “-“ between [total annual district cost] and [Total annual regional cost]. The correct formula is as follows:

[Total annual facility cost] + [total annual district cost] + [Total annual regional cost] + [National Admin Cost] + [National vaccine cost] = [Total annual national cost]

This has been corrected in the revised version of the manuscript (See Figure 1).

We confirmed that this formula is in line with the “Common Approach for the Costing and Financing Analyses of Routine Immunization and New Vaccine Introduction Costs (EPIC)”. Working Paper, Mimeo (2014) as well as with a publication of Brenzel et al. “Costs and Financing of Routine Immunization: Approach and Selected Findings of a Multi-Country Study (EPIC)”. Vaccine 2015, 33 Suppl 1, A13-20, doi:10.1016/j.vaccine.2014.12.066 (See Figure 2).

Reviewer 2 Report

Thank you for the opportunity to review this manuscript. In this paper, Gosset et al. report an analysis of both the economic and financial costs of introducing the hepatitis B birth dose vaccine in Senegal. This manuscript is presented, and the information here can be helpful to promote vaccine coverage in developing countries. Please see my specific comment stated below.

  1. Please explain the difference between financial cost and economic cost somewhere. This is common sense for certain readers, but I think it would be beneficial for others.
  2. Do you have any analysis or discussion of the cost-effectiveness of introducing this vaccine in a nation?

Author Response

1. Please explain the difference between financial cost and economic cost somewhere. This is common sense for certain readers, but I think it would be beneficial for others.

The difference between financial and economic costs has been clarified in the first paragraph of the subsection 2.3.1. p4 and in the third paragraph of the subsection 2.3.2. p6-7:

First paragraph of the subsection 2.3.1. p4 :

“Economic costs included a valuation of all inputs needed for the introduction and routine implementation of HepB-BD including valuation of time, supplies and equipment already existing in the EPI and used for HepB-BD (as the use of these resources represents an opportunity cost for the EPI) [16,17]. Financial costs were defined in terms of how much money was paid for the additional resources used for HepB-BD introduction and implementation (16). Accordingly, no financial cost was considered for activities carried out as part of the overall implementation of the EPI (such as vaccine storage and distribution, programme supervision and management, and waste management) unless additional resources were specifically required for HepB-BD introduction or implementation. Furthermore, capital costs were annualised over their expected lifetime using a discount rate of 3% in the economic approach and a straight-line depreciation was used in the financial approach.”

Third paragraph of the subsection 2.3.2. p6-7:

“Pre-introduction, initial training, social mobilization and HepB-BD vaccine administration were specific to the introduction and implementation of HepB-BD. Accordingly, their whole cost was attributed to HepB-BD both in the economic and financial approach (but the financial costs do not include labour time as no additional staff were hired). The rest of the activities were related to the implementation of the EPI as a whole. To estimate economic costs, we first estimated the EPI costs associated with each of these activities and then allocated a share of this cost to the HepB-BD implementation using the allocation criteria (Table 1). To estimate financial costs, we only considered expenditures related to the purchase of resources specifically required for HepB-BD introduction or implementation.”

2. Do you have any analysis or discussion of the cost-effectiveness of introducing this vaccine in a nation?

As suggested, the Discussion has been modified as follows (See p15):

“Our estimations of the economic costs in Senegal will also be especially useful to estimate and confirm findings from two studies suggesting the cost-effectiveness of introducing HepB-BD into sub-Saharan African EPI [28,29]”.
